# Long-Term Oncological Outcomes of Granulocyte Colony-Stimulating Factor (G-CSF) Treatment in Gastrointestinal Cancers: A Systematic Review and Meta-Analysis

**DOI:** 10.3390/cancers17081313

**Published:** 2025-04-14

**Authors:** Oliver Wedel Fischer, Tobias Freyberg Justesen, Dilara Seyma Gögenur, Michael Tvilling Madsen, Michael Bau Mortensen, Ismail Gögenur, Adile Orhan

**Affiliations:** 1Center for Surgical Science, Department of Surgery, Zealand University Hospital, 4600 Køge, Denmarkaor@regionsjaelland.dk (A.O.); 2Department of Surgery, Slagelse Sygehus, 4200 Slagelse, Denmark; 3Odense PIPAC Center, Odense University Hospital, 5000 Odense, Denmark; 4Odense Pancreas Center (OPAC), Odense University Hospital, 5000 Odense, Denmark; 5Department of Clinical Research, Faculty of Health Sciences, University of Southern Denmark, 5000 Odense, Denmark; 6Department of Surgery, Odense University Hospital, 5000 Odense, Denmark; 7Department of Clinical Medicine, University of Copenhagen, 2200 Copenhagen, Denmark

**Keywords:** G-CSF, filgrastim, peg-filgrastim, gastrointestinal cancer, survival, adverse events

## Abstract

This study investigated whether granulocyte-colony stimulating factor (G-CSF), a treatment that helps prevent infections during cancer therapy, could improve outcomes for patients with gastrointestinal (GI) cancers. We analyzed 13 studies with a total of 2673 patients. The use of G-CSF was associated with improved overall survival, although it did not have a clear effect on the length of time patients lived without their cancer worsening. Additionally, G-CSF use helped reduce neutropenia (bone marrow suppression), a common side effect of chemotherapy. While the evidence regarding G-CSF’s impact on survival was inconclusive, the reduction in neutropenia was supported by the data.

## 1. Introduction

Myeloid colony-stimulating factors such as (G-CSF) are widely utilized in patients with a diverse range of solid tumors undergoing myelosuppressive chemotherapy [1]. The use of G-CSF treatment has increased considerably since its invention. It has repeatedly been shown that the use of G-CSF reduces the risk of chemotherapy-induced neutropenia (CIN) and febrile neutropenia (FN) and considerably shorten the hospitalizations of patients [2,3]. The prophylactic use of G-CSF is generally thought to be a safe way to avoid potentially fatal side effects of myelosuppressive chemotherapy, particularly CIN and FN [2,3,4]. However, recent research also suggests that solid tumors may produce and secrete G-CSF [5,6]. Several tumor promoting mechanisms are potentially linked to G-CSF and its receptor, such as the increased proliferation and migration of cancer cells, tumor neo-angiogenesis, deprived adaptive immune responses through IFNγ and IL-17 dependent mechanisms and reduction in the infiltration of CD4^+^ and CD8^+^ T cells into the tumor microenvironment (TME) [5,6,7,8]. Especially in GI tumors, high plasma levels of G-CSF are associated with higher tumor stage, poor prognosis and overall survival (OS) [5,6,7]. Myeloid derived suppressor cells (MDCSs), which in recent research have been associated with reduced response to oncological treatments and rapid progression, have also been linked to high G-CSF plasma levels and G-CSF-producing tumors [9,10].

In GI cancers, the standard chemotherapy regimens (e.g., FOLFIRINOX, FOLFOX, FOLFIRI) are associated with bone marrow suppression and mucosal injury, increasing the risk of neutropenic enterocolitis (NE), which is a life-threatening condition accounting for approximately 5% of hospitalization after antineoplastic treatment [11]. Patients at risk of developing NE or patients who have been admitted to the hospital due to NE often receive G-CSF treatment to prevent mortality, which has previously been reported as being as high as 50% [11]. Thus, the use of G-CSF may be more frequent in patients with GI cancers compared to non-GI cancers. Furthermore, chemotherapy doses may often need to be adjusted or delayed due to neutropenia. The use of G-CSF as part of chemotherapy regimens could prevent dose reduction or delays, thereby improving treatment continuity. Thus, G-CSF holds an important role as a potential supportive care measure. These considerations emphasize the relevance of examining the impact of G-CSF on survival measures and adverse events (AEs) in GI cancers. Thus, the aim of this systematic review and meta-analysis was to investigate the association between G-CSF administration and oncological outcomes in GI cancers. Furthermore, this systematic review and meta-analysis aimed to investigate whether G-CSF administration in humans was associated with changes in circulating MDSC levels and changes in the density of MDSCs in the TME, specifically in GI-cancers.

## 2. Methods

### 2.1. Protocol

The protocol of this systematic review was composed in accordance with the PRISMA-P guidelines [12]. The protocol was submitted to PROSPERO on 2 February 2023 (CRD42023391556).

### 2.2. Search Strategy

A search of the literature was conducted on four different electronic data platforms: PubMed, Embase, Cochrane Library and Web of Science. The search strategy was based on the patient, intervention, comparison and outcome (PICO) process. Keywords and texts searches, in addition to medical subject headings and abbreviations, were utilized in the elaboration of the search strategy. No search restrictions were applied. The search strategy and an example of the search string is described in detail in Appendix A.

### 2.3. Selection Process and Data Management

Once the literature search was conducted, the acquired articles were imported to the management and screening tool Covidence (Covidence systematic review software, Veritas Health Innovation, Melbourne, Australia. Available at www.covidence.org, 9 April 2025). Duplicates were automatically detected by Covidence. The first author validated possible duplicates identified by Covidence. Additional duplicates were identified manually once the screening process commenced.

The screening process was conducted by two independent reviewers (OWF and DSG), who initially screened the title and abstract of articles for possible relevance based on the criteria of eligibility and then conducted the full text screening. The reviewers conferred with two independent supervisors (A.O. and T.F.J.) in cases of uncertainty of inclusion.

### 2.4. Eligibility Criteria

Study inclusion criteria included patients with GI cancers, including esophageal, gastro-esophageal junction, gastric, hepatic, intra- and extrahepatic bile duct, pancreatic (including ampullary tumors) and colorectal cancers. Patients had to be have been treated with myelosuppressive/myeloablative chemotherapy and been administered G-CSF. The G-CSF could be utilized as primary and secondary prophylaxis. There was no restriction regarding disease stage or prior surgical resection.

Reviews, letters, conference abstracts, commentaries, case reports, animal studies and studies of foreign languages which could not be translated, were all excluded. The full list of the eligibility criteria is available in Appendix A.

### 2.5. Prognostic Outcomes and Safety

The primary outcome was the association between G-CSF use and oncological outcomes. Outcomes of interest were OS, progression-free survival (PFS), disease-free survival (DFS), cancer-specific survival (CSS) and recurrence. The definitions of the aforementioned outcomes are based on the National Cancer Institute’s (NCI) Dictionary of Cancer Terms [13]. The safety of G-CSF treatment was assessed by grade III/IV neutropenia, as defined by the National Cancer Institute Common Terminology Criteria for Adverse Events (NCI CTCAE v5.0) [14].

### 2.6. Data Extraction

A standardized data extraction form was made for the extraction of the following data: first author, year of publication, study design, country of origin for study population, type of cancer, sample size, chemotherapy regimen, type of G-CSF and investigated prognostic outcomes.

Hazard ratios (HRs) and their 95% confidence intervals (CIs) for the outcomes of interests were extracted. In cases where studies solely provided Kaplan–Meier (KM) plots without presenting HR values, the HR was approximated from the KM plots using the Digitizelt desktop application (Digitizelt 2.6) as outlined by Tierney et al. [15].

Corresponding authors of potentially eligible studies with missing data were contacted by email. The studies were not included in the meta-analysis if no response was obtained.

### 2.7. Data Synthesis and Subgroups

Time-to-event data meta-analyses were performed in R-Studio, version 4.1.3, using a random effects model, and this was accomplished by utilizing the “meta” and “metafor” packages. The Sidik–Jonkman estimator for tau2 was applied with Hartung–Knapp adjustment in the random effect model [16]. Prediction intervals were calculated to provide estimates of the expected effect size of future studies based on current evidence [17,18]. Heterogeneity was assessed by Chi^2^ testing and I^2^ statistics. In the meta-analysis performed on event data (AEs) the effect estimate was relative risk (RR), and the inverse variance method was applied in a random effects model. Otherwise, similar settings for the analysis were applied. Missing data were handled in accordance with the recommended approaches outlined in the Cochrane Handbook for Systematic Reviews of Interventions, version 6.4.

Meta-analyses were presented separately depending on outcomes of interest. If applicable, subgroup analyses based on the study design (randomized clinical trial (RCT), or cohort study) or cancer type (gastric cancer, pancreatic cancer or colorectal cancer) were conducted.

### 2.8. Study Quality Assessment and Risk of Bias

In order to assess the quality of the included studies, the Newcastle–Ottawa Quality Assessment Scale (NOS) was applied for cohort studies. This tool was used to score studies in the following categories: selection, comparability and outcome. The score from the NOS was converted via the Agency for Healthcare Research and Quality (AHRQ) standards into either good, fair or poor quality. The NOS quality assessment is shown in Appendix A.

For RCTs, the Cochrane Risk of Bias Tool for Randomized Controlled Trials (RoB) was utilized. This tool was used to evaluate the studies in the following domains: random sequence generation, allocation concealment, selective reporting, other bias, blinding of participants and personnel, blinding of outcome assessment and incomplete data. In each of the aforementioned categories, the study was awarded either low, high or unknown risk of bias. The score from the RoB was converted via the AHRQ standards into either good, fair or poor quality. The RoB quality assessment is shown in Appendix A.

Publication bias of the different studies and their associated outcomes were visually inspected through funnel plots and statistically assessed through Egger’s test of the intercept, in which a *p* value < 0.05 was considered to indicate the presence of publication bias.

### 2.9. Confidence in Cumulative Evidence

The quality of the cumulative evidence was evaluated using a modified version of the GRADE approach [19]. The modified approach was used as this meta-analysis included both RCTs and cohort studies.

## 3. Results

### 3.1. Search Results

The literature search in the four databases were conducted in November 2022, yielding 2955 studies. After the removal of duplicates, a total of 2452 studies were eligible for title and abstract screening. Consequently, 251 studies were then included for full-text screening. After the full-text screening process, 201 studies were excluded, ultimately leaving 50 studies for final inclusion in this review. During full-text screening, 37 studies [20,21,22,23,24,25,26,27,28,29,30,31,32,33,34,35,36,37,38,39,40,41,42,43,44,45,46,47,48,49,50,51,52,53,54,55,56] lacked a comparator group, leaving 13 studies [57,58,59,60,61,62,63,64,65,66,67,68,69] for inclusion in the final meta-analysis (Appendix A).

The inclusion process and reasons for exclusion are displayed in the PRISMA flow-chart (Figure 1). Of the 201 full-text reviewed excluded studies, 75 studies had an unsystematic or inconsistent use of G-CSF, 43 studies were abstracts or conference posters, 23 studies had unrelated outcomes, 21 studies had no data available for extraction, 13 studies had no accessible full-texts, 12 studies reported results on non-GI cancers, 6 studies had wrong study-designs, 5 studies did not cover the intervention of interest and 3 studies were reviews.

### 3.2. Study Characteristics

In total, 13 studies with a total of 2673 patients were included in this review and meta-analysis [57,58,59,60,61,62,63,64,65,66,67,68,69]. Of the included studies, six were conducted in North America, five in Asia, two in Europe and one in South America. All the studies were written in English and were published between 2000 and 2022. Eight studies were RCTs, four were retrospective cohort studies and one was non-RCT. The distribution of cancer examined by the studies were grouped as follows: five studies reported on colorectal cancer, four on pancreatic cancer, three on gastric cancer and one on esophageal cancer. All of the studies treated one group of the study population with G-CSF, and the other group with either placebo or no G-CSF in the comparator group. The specific G-CSF drugs utilized in these studies were filgrastim (four studies), pegfilgrastim (four studies) or a mix of either filgrastim or pegfilgrastim (two studies). Three studies did not specify which G-CSF drug was used. The investigated chemotherapy regimens were quite diverse and are visualized in Table 1, alongside the abovementioned study characteristics. The follow-up duration varied from 20 to 84 months in the included studies. No studies reported data on CSS, DFS or recurrence. None of the studies examined MDSCs in relation to G-CSF use and survival outcomes in humans. Thus, no data on MDSCs in relation to the outcomes of interest are reported.

### 3.3. Overall Survival

Ten studies reported data on OS [57,59,60,61,62,64,65,67,68,69]. Across all types of included cancers and study designs, a significant improvement on OS amongst G-CSF-treated patients compared to those with placebo/no prophylaxis were found (HR 0.72, 95% CI: 0.56–0.91, *I*^2^: 54%, Figure 2A). However, as evident from the prediction interval (95% CI: 0.33–1.53), other studies cannot confirm these findings.

The subgroup analyses of OS in RCTs (HR 0.81, 95% CI: 0.58–1.13, *I*^2^: 48%, Figure 2B) versus observational studies (HR 0.62, 95% CI: 0.37–1.02, *I*^2^: 35%) revealed a trend towards improved OS in both study designs. No significant difference was found between subgroup study design (*p* = 0.212).

In the subgroup analysis of OS according to the different GI cancer examined, the results on OS in colorectal cancer (HR 0.93, 95% CI: 0.85–1.01, *I*^2^: 0%), gastric cancer (HR 0.75, 95% CI: 0.03–21.14, *I*^2^: 73%) and pancreatic cancer (HR 0.58, 95% CI: 0.29–1.14, *I*^2^: 43%) revealed no significant results (Appendix A). Inter-subgroup tests of differences revealed a significant result (*p* = 0.023), suggesting different effects of G-CSF across cancer types.

### 3.4. Progression-Free Survival

Eight studies reported data on PFS [57,59,61,62,64,65,67,69]. In the pooled analysis, which included all types of GI cancers and study designs, the result showed a potential improvement of PFS in patients receiving G-CSF prophylaxis; however, this was not significant (HR 0.74, 95% CI: 0.51–1.08, *I*^2^: 73%, Figure 3A). Also, due to the high heterogeneity between included studies, the prediction interval was correspondingly wide 0.26–2.16 95% CI.

The subgroup analysis of the RCTs reporting on PFS showed a non-significant impact on PFS (HR 0.97, 95% CI: 0.82–1.16, *I*^2^: 0%). Interestingly, the subgroup analysis of the retrospective cohorts displayed significant improvement in PFS in patients receiving G-CSF prophylaxis (HR 0.50, 95% CI: 0.32–0.77, *I*^2^: 0%, Figure 3B). This difference in subgroups was statistically significant *p* < 0.001, indicating overall no effect on PFS in RCTs. This raises concern regarding the potential of bias in the non-randomized trials. Thus, the results should be interpreted with caution and further RCTs are needed to verify the observed effects in this meta-analysis.

Subgroup analysis on studies investigating colorectal cancer (HR 0.91, 95% CI: 0.68–1.21, *I*^2^: 0%), gastric cancer (HR 1.08, 95% CI: 0.20–5.81, *I*^2^: 0 %) and pancreatic cancer (HR 0.47, 95% CI: 0.02–13.48, *I*^2^: 28%) were non-significant. However, significant subgroups differences were present *p* = 0.0017 (Appendix A).

### 3.5. Adverse Events

In total, eight studies reported data on AEs [57,59,61,62,64,65,67,68]. A significant decrease in the risk of grade III/IV neutropenia was seen in patients receiving G-CSF prophylaxis (RR 0.46, 95% CI: 0.28–0.77, *I*^2^: 72%, Figure 4A), although the prediction interval still contained the null effect 95% CI 0.09–2.39. The significantly lower RR was also seen in the subgroup analysis of the RCTs (RR 0.37, 95% CI: 0.15–0.94, *I*^2^: 79%, Figure 4B). However, in the retrospective cohorts, a non-significant association was found (RR 0.57, 95% CI: 0.26–1.29, *I*^2^: 59%). The test of inter-subgroup differences were non-significant (*p* = 0.349). Furthermore, the heterogeneity observed in these analyses was concerningly high.

Subgroup analysis on the GI cancer location (Appendix A) revealed an association favoring G-CSF prophylaxis in colorectal cancer (RR 0.24, 95% CI: 0.06–0.91, *I*^2^: 42%). Subgroup analyses on gastric cancer (RR 0.73, 95% CI: 0.13–4.02, *I*^2^: 0%) and pancreatic cancer (RR 0.58, 95% CI: 0.18–1.85, *I*^2^: 0%) showed no significant association between G-CSF use and AEs. The test of inter-subgroup differences were significant (*p* = 0.0097).

### 3.6. Myeloid-Derived Suppressor Cells (MDSCs)

Due to the immunosuppressive properties of myeloid-derived suppressor cells (MDSCs), it was also of interest to evaluate studies investigating the association between G-CSF use and the level of MDSCs. However, the search strategy revealed no studies on this matter.

### 3.7. Study Quality and Risk of Bias

The NOS was used to assess risk of bias in cohort studies and revealed that all the included studies were of good quality, with the lowest number of awarded stars being six. The risk of bias assessment using NOS is available in Appendix A.

The RoB was used on the RCTs. Four of the seven studies were assessed as being of good quality with low risk of bias, one study was assessed as fair quality and two studies were assessed as poor quality. The two studies of poor quality were due to the basis of unclear allocation concealment, unclear blinding of participants and personnel and/or unclear blinding of outcomes. There is a likely risk of the outcomes of these two studies being biased based on this assessment. The risk of bias assessment using the RoB 2 tool is available in Appendix A.

### 3.8. Certainty in Evidence

The cumulative evidence obtained for the outcomes of interest was of differing quality. The quality was high for the evidence obtained for G-CSF use and AE. The quality of evidence was low in the OS group and of moderate quality in the PFS group. The evaluated quality of the evidence for each outcome of interest is available in Appendix A, and the grading is summarized in detail there.

## 4. Discussion

The results of this meta-analysis revealed a significantly longer OS in patients with GI cancers treated with G-CSF. Patients undergoing myelotoxic/myeloablative chemotherapy regimen for a GI cancer, could therefore potentially gain an increased OS with the addition of G-CSF prophylaxis (HR 0.72, 95% CI: 0.56–0.91, *I*^2^: 54%). We did not find a significant impact of G-CSF prophylaxis on PFS; however, the subgroup analysis of retrospective cohort studies did reveal an association favoring G-CSF prophylaxis.

In terms of chemotherapy-induced neutropenia grade III/IV, the addition of G-CSF prophylaxis in a chemotherapy regimen showed significantly reduced AEs in patients. The incorporation of G-CSF prophylaxis drastically decreased the rate of neutropenia (RR 0.46, 95% CI: 0.28–0.77, *I*^2^: 72%). These results are in accordance with data from prior studies [2,3,4]. 

This review had multiple limitations, the most prominent being the limited published literature with only a few large RCTs and heterogeneous study populations. A multitude of oncological studies examined the effects of different chemotherapy regimens, and a large portion of these utilized G-CSF prophylaxis. Nevertheless, most of these studies failed to report the number of patients treated with G-CSF, nor were the patients in these studies separated from the non-G-CSF-treated patients in the data analysis (Appendix A). This lack of subgroup analyses of G-CSF-treated patients could potentially be a significant confounder in retrospective cohorts [59]. Another limitation of this study was the heterogeneity in the chemotherapy regimens used in the included studies. The studies had a broad range of chemotherapy regimens with different administration, dosing, toxicity and side-effect profiles, which ultimately may also impact OS, PFS and AEs. Also, none of the studies included patients treated with a combination of chemo- and radiation therapy (CRT) or radiotherapy (RT) alone, although this approach may be used in esophageal, rectal and pancreatic cancer. The potential radio mitigating effect of G-CSF in patients with GI cancer is unknown but may also be explored in future studies [70]. Thus, subgroup analyses on the type of GI cancer examined were conducted to reduce heterogeneity.

Prior systematic reviews and meta-analyses assessing the safety profile of G-CSF, have concluded that G-CSF can be administered without a significant risk of affecting tumor response or OS [2,3,4]. Safety studies are the backbone of the informational material on toxicity in the current European Society for Medical Oncology (ESMO) clinical practice guidelines for administration of G-CSF [71]. Nonetheless, these aforementioned reviews and meta-analyses consist mainly of studies with hematological malignancies and solid tumors such as breast, lung and ovarian cancers. Few of the included studies are on GI tumors, and those that are included are small in comparison and do not include subgroup analyses. Thus, important information regarding G-CSF impact on exclusively GI cancers are overlooked. To the best of our knowledge, this meta-analysis is the first to focus solely on GI cancers.

The current ESMO guidelines for administration of G-CSF states that G-CSF prophylaxis is indicated when the risk of febrile neutropenia (FN) is greater than 20% and must be considered when the risk of FN is 10–20%, depending on age, comorbidities and specific chemotherapy regime [71]. Nonetheless, when patients are assessed as being at high risk of developing FN and are therefore administered G-CSF, they are more likely to receive G-CSF in the event of future recurrence or a change in chemotherapy regimen [72]. Thus, patients are at risk of entering a cycle of continuous G-CSF prophylaxis even though conditions for administration are not fulfilled.

With regard to the included studies in this review and the clinical practice guidelines for G-CSF, the survival benefit associated with G-CSF administration may be secondary to increased chemotherapy doses. Studies have found that patients undergoing G-CSF supported myelotoxic chemotherapy often are subjected to a larger degree of high dose chemotherapy [59,73]. A systematic review and meta-analysis of 59 RCTs of patients with non-GI cancers administered chemotherapy and G-CSF or placebo, found that G-CSF prophylaxis was associated with higher dose intensity (DI) and a reduction in risk of all-cause mortality (RR = 0.93; 95% CI: 0.90–0.96; *p* < 0.001). The greatest impact on survival was observed in the studies with the highest DI, or dose-dense schedules [74]. The beneficial effects of G-CSF on OS, which are observed in the majority of the studies, might therefore be secondary to the amplification of chemotherapy doses.

The aim of this systematic review and meta-analysis was to examine both the neutropenia preventing properties of G-CSF and the drug’s potential role in recruiting MDSCs. However, no clinical trials examining the role of G-CSF on MDSC levels were found during the literature search. Thus, we could not deduce anything about these considerations about G-CSF. Pre-clinical trials have suggested that G-CSF treatment may cause the upregulation of G-CSFR [6], which is associated with tumor secretion of G-CSF, lower OS, advanced stages and more aggressive metastasis pattern [5,7]. It has been found that, especially in GI cancers, there is a higher prevalence of these tumors with an upregulation of G-CSFR as well as a tumor secretion of G-CSF [6]. In this regard, it is unknown what consequences prophylactic treatment with G-CSF imposes in cancer patients. In vitro studies, however, have identified gastric and colon carcinoma cells to be susceptible to high G-CSF doses, resulting in cancer development in terms of increased cell proliferation, migration and invasion [75]. In vitro studies also show that G-CSF can mitigate the increased recruitment and mobilization of MDSCs, inducing VEGF-independent angiogenesis with following increased resistance to anti-VEGF therapeutics [9,10]. With their strong immunosuppressive functions, MDSCs can decrease T cell function. A recent systematic review and meta-analysis of 40 studies showed that MDSCs and their subtypes all are associated with poor prognosis in terms of OS in patients with solid tumors [76].

How and if G-CSF prophylaxis could be associated with tumor progression, MDSC mobilization and recruitment and, ultimately, worse prognosis is simply not known in humans. The abovementioned studies are mostly hypothesis-generating. It was also the aim of this review to uncover studies examining G-CSF treatment in association with circulating MDSC levels. However, to the best of our knowledge, no such studies exist. This emphasizes the need for further research on the subject.

## 5. Conclusions

Based on this meta-analysis, despite the limited literature, G-CSF treatment is associated with a reduction in the risk of neutropenia grade III/IV. However, the heterogeneity is high. The meta-analysis also suggested a favorable OS among G-CSF-treated patients with GI cancer, but the predicting interval suggests that future studies on this subject may not confirm this benefit in OS. In addition, PFS was unaffected in patients treated with G-CSF, and the subgroup analysis based on the study design revealed no observed effect in RCTs.

## Figures and Tables

**Figure 1 cancers-17-01313-f001:**
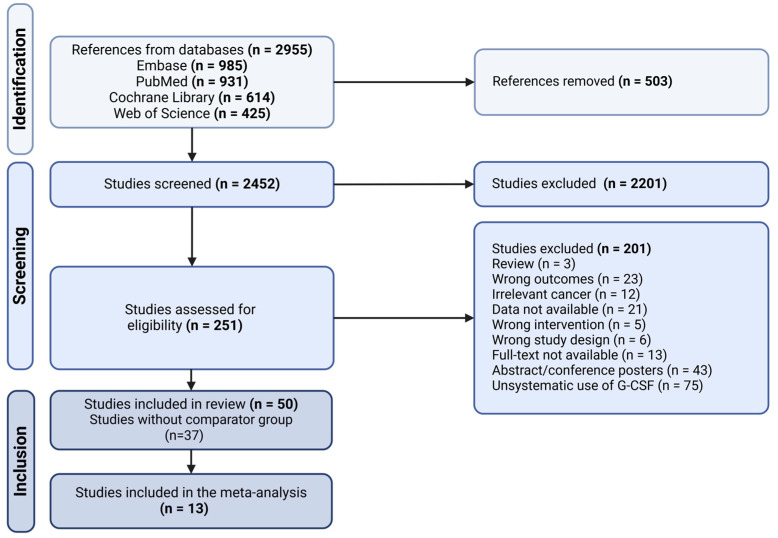
PRISMA flow-chart. n represents the sample size. Created in BioRender. Fischer, O. https://BioRender.com/cp76rte (accessed on 23 March 2025).

**Figure 2 cancers-17-01313-f002:**
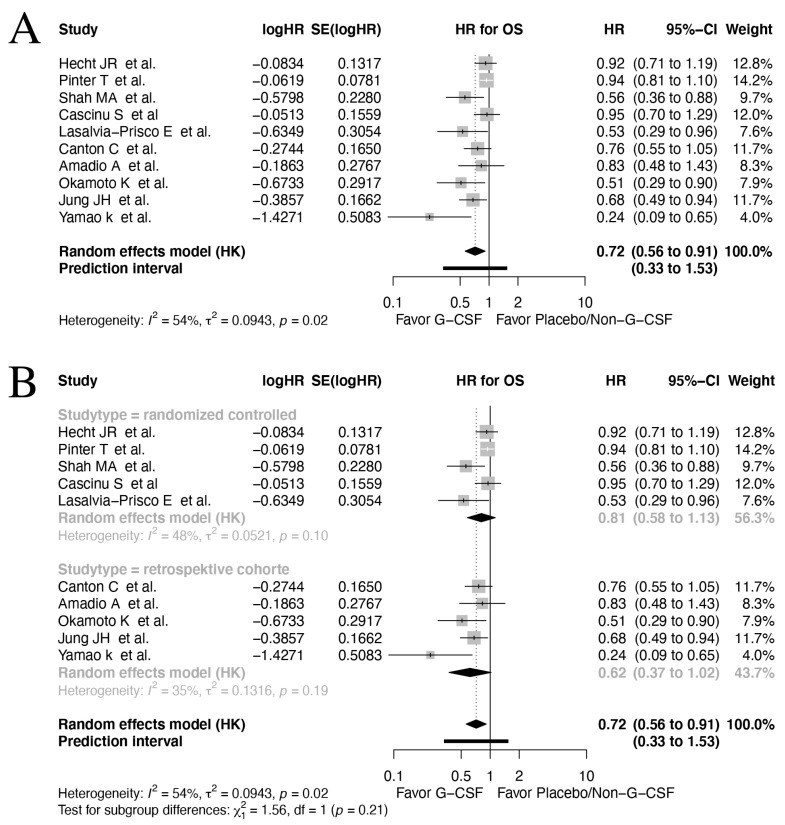
Forrest plot on all studies reporting OS [57,59,60,61,62,64,65,67,68,69]. (**A**) and subgroup analysis regarding study design (**B**) applying a random effect model on time-to-event data. CI = confidence interval; HR = hazard ratio; OS = overall survival; TE = ln(HR); seTE = standard error for ln(HR).

**Figure 3 cancers-17-01313-f003:**
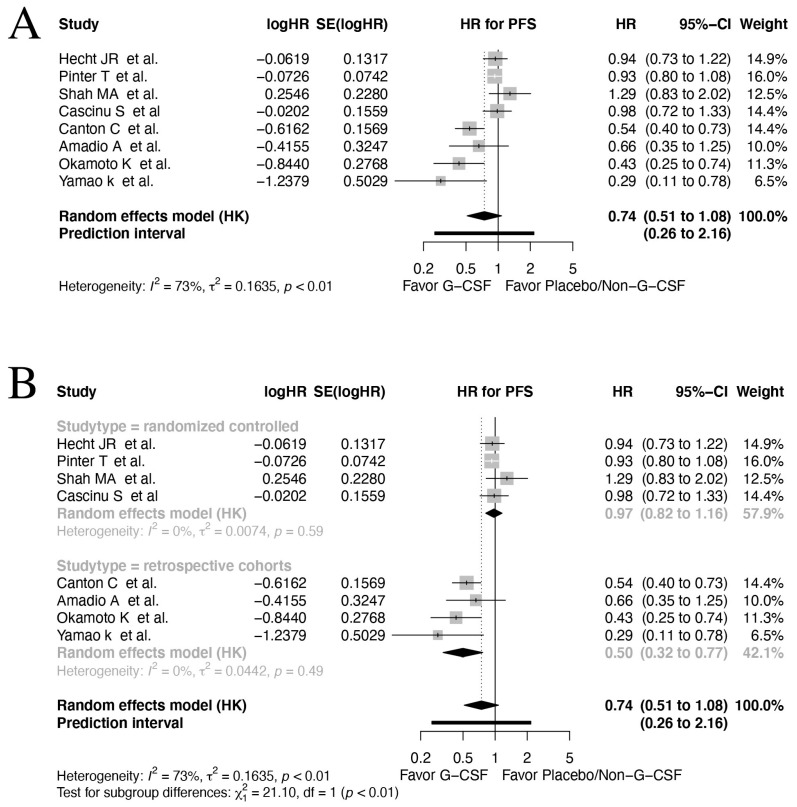
Forrest plot on all studies reporting PFS [57,59,61,62,64,65,67,69]. (**A**) and subgroup analysis regarding study design (**B**) applying a random effect model on time-to-event data. CI = confidence interval; HR = hazard ratio; PFS = Progression-free survival; TE = ln(HR); seTE = standard error for ln(HR).

**Figure 4 cancers-17-01313-f004:**
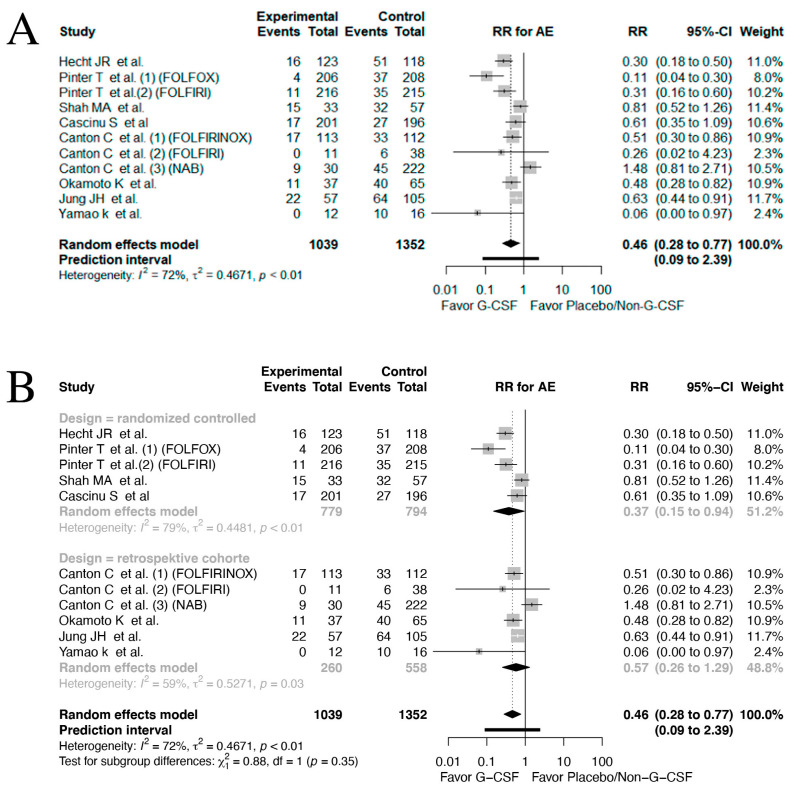
Forrest plot on all studies reporting AE [57,59,61,62,64,65,67,68]. (**A**) and subgroup analysis regarding study design (**B**) applying a random effect model on time-to-event data. CI = confidence interval; HR = hazard-ratio; AE = adverse event(s); TE = ln(HR); seTE = standard error for ln(HR).

**Table 1 cancers-17-01313-t001:** Study characteristics of the 13 included studies.

Study	Country	Study Design	Cancer	*n*	Chemotherapy	G-CSF	Outcomes	Follow-Up (Months)
Wadler S et al. 2002 [63]	US	RCT	Gastric	23	FHIG ^a^/AD ^b^	Filgrastim	OS, PFS	Not specified
Cascinu S et al. 2007 [61]	Italy	RCT	Gastric	400	Cis+5-FU+epi ^c^/5-FU/LV ^d^	Filgrastim	OS, PFS, AE	60
Hecht JR et al. 2010 [65]	US	RCT	Colorectal	252	FOLFOX ^e^/FOLFIRI ^f^/FOIL ^g^	Pegfilgrastim	OS, PFS, AE	24
Lasalvia-Prisco E et al. 2012 [60]	Uruguay	RCT	Pancreatic	60	GCD ^h^/GCD + various support ^i^	Not specified	OS,	24
Shah MA et al. 2015 [62]	US	RCT	Gastric	90	DCF ^j^	Filgrastim	PFS, OS, AE	42
Chen J et al. 2017 [66]	China	RCT	Colorectal	100	Mixed ^k^	Not specified	AE	Not specified
Pinter T et al. 2017 [64]	US	RCT	Colorectal	845	FOLFOX ^e^/FOLFIRI ^f^	Pegfilgrastim	PFS, OS, AE	58
Pitot HC et al. 2000 [58]	US	Cohort	Colorectal	48	9AC ^l^	Not specified	PFS, OS, AE	30
Amadio A et al. 2014 [69]	Canada	Non-RCT	Colorectal	62	FOLFIRI ^f^	Filgrastim/ Pegfilgrastim	OS, PFS	60
Yamao k et al. 2019 [57]	Japan	Cohort	Pancreatic	28	FOLFIRINOX ^m^	Pegfilgrastim	PFS, OS	20
Jung JH et al. 2020 [68]	South Korea	Cohort	Pancreatic	165	FOLFIRINOX ^m^	Filgrastim/ Pegfilgrastim	OS, AE	84
Okamoto K et al. 2022 [67]	Japan	Cohort	Oesophageal	102	DCF ^j^	Pegfilgrastim	PFS	72
Canton C et al. 2022 [59]	France	Cohort	Pancreatic	498	Mixed ^k^	Not specified	OS, PFS	Not specified

OS: overall survival; PFS: progression-free survival; AE: adverse events; G-CSF: granulocyte-colony stimulating factor; RCT: randomized controlled trial. a: Fluorouracil + hydroxyurea + IFNα2; b: doxorubicin + docetaxel; c: cisplatin + fluorouracil + epidoxorubicin + 6S-LV; d: fluorouracil + 5S-LV; e: leucovorin + fluorouracil + oxaliplatin; f: leucovorin + fluorouracil + irinotecan; g: fluorouracil + leucovorin + oxaliplatin + irinotecan; h: gemcitabine + cisplatin + docetaxel; i: cyclophosphamide + celecoxib + sulfhydryl; j: docetaxel + cisplatin + fluorouracil; k: various chemotherapy regimens was utilized; l: doxorubicin + cyclophosphamide; m: leucovorin + fluorouracil + irinotecan + oxaliplatin.

## Data Availability

Available by request to corresponding author (Fischer, O.W) or in Appendix A.

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
