# Peer review of "Long-Term Oncological Outcomes of Granulocyte Colony-Stimulating Factor (G-CSF) Treatment in Gastrointestinal Cancers: A Systematic Review and Meta-Analysis"

_cancers, 2025, doi:10.3390/cancers17081313_

Round 1
Reviewer 1 Report
Comments and Suggestions for Authors
In this study the authors address a significant gap in oncological research by systematically reviewing the effects of G-CSF prophylaxis on gastrointestinal (GI) cancers. The novelty of this meta-analysis is its exclusive focus on GI cancers, which have not been thoroughly studied in previous systematic reviews. The study follows PRISMA guidelines and includes robust methodological elements such as risk-of-bias assessment (RoB and NOS), subgroup analyses, and prediction intervals, which enhance credibility. The use of time-to-event random-effects meta-analysis and tools such as the Hartung-Knapp adjustment and Sidik-Jonkman estimator ensures a well-calibrated statistical approach. Overall, this study highlights the impact of G-CSF on overall survival (OS) and progression-free survival (PFS), while also considering adverse events (AEs), which is useful for clinical decision-making.
Nether less, I noticed some weaknesses in the paper:
First of all, the studies analyzed vary widely in chemotherapy regimens, patient populations, and cancer types, leading to potential confounding effects. While OS improvement is reported (HR 0.72, 95% CI: 0.56–0.91), the prediction interval (95% CI: 0.33–1.53) suggests that future studies might not confirm this benefit. Also, PFS results are conflicting: no overall effect in RCTs, but a significant effect in retrospective cohorts (HR 0.50, 95% CI: 0.32–0.77), raising concerns about potential bias in non-randomized studies. I think that the quality of evidence for OS is rated as low, and for PFS, it is moderate. This weakens the strength of conclusions.
On my opinion, the paper hypothesizes about G-CSF’s role in immunosuppression and tumor progression through myeloid-derived suppressor cells (MDSCs), but no included studies provide data on this aspect.
For the reasons above, I would suggest for example to emphasize more clearly the distinction between RCTs and retrospective studies. The statement "G-CSF prophylaxis significantly reduced neutropenia grade III/IV rates (RR 0.46, 95% CI: 0.28–0.77, I² = 72%)" should acknowledge the high heterogeneity. It would strengthen the credibility of the meta-analysis to include a sentence addressing potential bias from lower-quality studies.
In conclusion, I think that the study is well-conducted, methodologically sound, and provides useful insights into G-CSF use in GI cancers. However, it faces significant heterogeneity, potential publication bias, and varying quality of evidence. To clarify the limitations of the results, particularly in OS and PFS, will improve the paper’s credibility. Addressing these concerns in the results section and adding cautious interpretations will make the conclusions more robust.
Author Response
Reviewer 1:
In this study the authors address a significant gap in oncological research by systematically reviewing the effects of G-CSF prophylaxis on gastrointestinal (GI) cancers. The novelty of this meta-analysis is its exclusive focus on GI cancers, which have not been thoroughly studied in previous systematic reviews. The study follows PRISMA guidelines and includes robust methodological elements such as risk-of-bias assessment (RoB and NOS), subgroup analyses, and prediction intervals, which enhance credibility. The use of time-to-event random-effects meta-analysis and tools such as the Hartung-Knapp adjustment and Sidik-Jonkman estimator ensures a well-calibrated statistical approach. Overall, this study highlights the impact of G-CSF on overall survival (OS) and progression-free survival (PFS), while also considering adverse events (AEs), which is useful for clinical decision-making.
Response:
We deeply thank the reviewer for the positive feedback and for summarizing the work in detail. We have addressed all the reviewer’s suggestions below.
Nether less, I noticed some weaknesses in the paper:
First of all, the studies analyzed vary widely in chemotherapy regimens, patient populations, and cancer types, leading to potential confounding effects. While OS improvement is reported (HR 0.72, 95% CI: 0.56–0.91), the prediction interval (95% CI: 0.33–1.53) suggests that future studies might not confirm this benefit. Also, PFS results are conflicting: no overall effect in RCTs, but a significant effect in retrospective cohorts (HR 0.50, 95% CI: 0.32–0.77), raising concerns about potential bias in non-randomized studies. I think that the quality of evidence for OS is rated as low, and for PFS, it is moderate. This weakens the strength of conclusions.
Response:
We thank the reviewer for the comments. We completely agree that the prediction interval for the reported HR for OS suggest that this benefit cannot be confirmed in future studies. We have further emphasized this in the results section at “3.3 Overall Survival” on page 7, line 224-225.
Regarding the results on PFS, we agreed with the reviewer that the observed no overall effect in RCTs is a limitation and indicates a possibility of bias. We have now specified this limitation and weakness of the data at the “3.4 Progression-free survival” section on page 8, lines 251-254.
Furthermore, we have now rephrased the conclusion of this manuscript, highlighting the limitations and that the data should be interpreted with caution. The included sentences can be found on page 13, line 288-392.
On my opinion, the paper hypothesizes about G-CSF’s role in immunosuppression and tumor progression through myeloid-derived suppressor cells (MDSCs), but no included studies provide data on this aspect.
Response:
We thank the reviewer for highlighting this issue. Unfortunately, the search on multiple databases revealed no studies providing data on the association between MDSCs and G-CSF (pegfilgrastim). Thus, no studies were included on this. We have now added a sub-heading in the “Results” section (section 3.6 at page 10, line 280-284) emphasizing the interest in looking at this relation between G-CSF and MDSCs but that no studies on this matter were found during the literature search.
For the reasons above, I would suggest for example to emphasize more clearly the distinction between RCTs and retrospective studies. The statement "G-CSF prophylaxis significantly reduced neutropenia grade III/IV rates (RR 0.46, 95% CI: 0.28–0.77, I² = 72%)" should acknowledge the high heterogeneity. It would strengthen the credibility of the meta-analysis to include a sentence addressing potential bias from lower-quality studies.
Response:
We again thank the reviewer for the suggestions. We have now included a sentence acknowledging the high heterogeneity in the results regarding adverse events. The changes can be found on page 9, line 271-272. Furthermore, we have also emphasized the high heterogeneity in the conclusion section on page 13, line 389-390.
In conclusion, I think that the study is well-conducted, methodologically sound, and provides useful insights into G-CSF use in GI cancers. However, it faces significant heterogeneity, potential publication bias, and varying quality of evidence. To clarify the limitations of the results, particularly in OS and PFS, will improve the paper’s credibility. Addressing these concerns in the results section and adding cautious interpretations will make the conclusions more robust.
Response:
We deeply appreciate the suggestions from the reviewer and thank the reviewer for the positive feedback as well. We have now addressed the limitations and emphasized the need for carefulness when interpreting the results. We have addressed the concerns regarding the data generated on both OS, PFS and on Adverse Events in both the results sections and in the conclusion of the paper. We believe the changes suggested by the reviewer have made the paper more credible.
Reviewer 2 Report
Comments and Suggestions for Authors
This paper is a metaanlaysis of GSF use showing some evidence of improved OS. The paper does a good job in managing expectations given limited data and does not try to over interpret the results. The methodology is sound.
My recommendations are more to do with formatting. The PRISMA table is hard to read and should be reformatted for readablity, espeically the study excluded box which is cut off. All forest plot figures are of poor resolution - please revise.
Author Response
Reviewer 2:
This paper is a metaanlaysis of GSF use showing some evidence of improved OS. The paper does a good job in managing expectations given limited data and does not try to over interpret the results. The methodology is sound.
My recommendations are more to do with formatting. The PRISMA table is hard to read and should be reformatted for readablity, espeically the study excluded box which is cut off. All forest plot figures are of poor resolution - please revise.
Response:
We thank the reviewer for the positive feedback and for the recommendations regarding formatting. We have now improved the quality of the PRISMA table and optimized the quality of the forest plots as well.
Reviewer 3 Report
Comments and Suggestions for Authors
Fisher et al evaluated the G-CSF treatment outcome in GI cancer patients. They surveyed the literature and used meta-analysis of 2673 (G-CSF treated) patients with to assess the Overall survival (OS), progression free survival (PFS) and adverse events(AE). They found that G-CSF has positive association with OS but not with PFS. They concluded that G-CSF provides reduction of neutropenia in in advanced stage (Grade III/IV) GI cancer patients.
However, the paper is not edited properly and contains numerous mistakes that need to be corrected. Some examples are below:
- Line 51, the sentence stating “risk of developing chemotherapy” should be re-written or modified.
- Introduction does not properly convey the exact reason for the study.
- In line 56-65, authors stated that solid tumors produce G-CSF and it helps in progression of tumor. In GI, higher G-CSF in plasma worsens the treatment. But in line 70-72, they stated that G-CSF treatment prevents mortalities with NE in GI. These two statements have opposite meanings that should be changed.
- In introduction, author should explain “OS” (overall survival) and “PFS” progression free survival
- Line 61, TNM should be expanded.
- Line 32, 2457 studies are included for eligibility. But the PRISMA chart (Figure 1) shows 2452 studies qualified. These numbers should be corrected.
- Line 188, RCT should be expanded in first appearance.
- Authors used many abbreviations that are not expanded, such as CSS (line 198), MDSCs (line 199) etc.
- In Figure 1A and B, authors obtained HR 0.72 in both cases. In fig 1B, the test of subgroups has a p value 0.21, signifies no difference. Whereas, in 1A, authors do not show any p-value (except heterogenicity) to compare or confirm. Why HR 0.72 should be favorable for G-CSF treatment? Is there any range of HR that should be considered as favorable or nonfavorable? These should be explained.
- Similarly in PFS, why HR 0.74 is not a significant improvement? Then authors should write no significant improvement in PFS over OS (line 233).
- Line 254, RR should be expanded.
- Line 272, NOS should be expanded. RoB should be expanded (line 275)
Comments on the Quality of English Language
In the introduction, authors' intentions are not properly expressed. Also, there are several grammatical mistakes in the manuscript.
Author Response
Reviewer 3:
Fisher et al evaluated the G-CSF treatment outcome in GI cancer patients. They surveyed the literature and used meta-analysis of 2673 (G-CSF treated) patients with to assess the Overall survival (OS), progression free survival (PFS) and adverse events(AE). They found that G-CSF has positive association with OS but not with PFS. They concluded that G-CSF provides reduction of neutropenia in in advanced stage (Grade III/IV) GI cancer patients.
However, the paper is not edited properly and contains numerous mistakes that need to be corrected. Some examples are below:
- Line 51, the sentence stating “risk of developing chemotherapy” should be re-written or modified.
Response:
We thank the reviewer for the thorough feedback on the manuscript. We are sorry for the mistake. The sentence has now been rewritten.
- Introduction does not properly convey the exact reason for the study.
Response:
We thank the reviewer. We have now further emphasized the reason for conveying this systematic review and meta-analysis in the “Introduction” section on page 2, line 82-86.
- In line 56-65, authors stated that solid tumors produce G-CSF and it helps in progression of tumor. In GI, higher G-CSF in plasma worsens the treatment. But in line 70-72, they stated that G-CSF treatment prevents mortalities with NE in GI. These two statements have opposite meanings that should be changed.
Response:
We thank the reviewer for the suggestion. Ideally, we wanted to elaborate both the NE and general neutropenia preventing properties of G-CSF/Peg-filgrastrim and the drugs potential role in recruiting MDSCs. However, no clinical trials examining the role of G-CSF on MDSC levels were found during the literature search. Thus, we could not deduce anything about these considerations about G-CSF. Nonetheless, we still found it relevant to include and introduce in the background to emphasize the need for future research on this matter.
- In introduction, author should explain “OS” (overall survival) and “PFS” progression free survival
Response:
We thank the reviewer for noticing this. We have now included what the abbreviation stands for in the introduction section at page 2, line 70 and in the methods section at page 3, line 123.
- Line 61, TNM should be expanded.
Response:
TNM has now been changed in the sentence on page 2, line 70.
- Line 32, 2457 studies are included for eligibility. But the PRISMA chart (Figure 1) shows 2452 studies qualified. These numbers should be corrected.
Response:
The numbers have now been corrected so that it is consistent between the PRISMA chart and in-text.
- Line 188, RCT should be expanded in first appearance.
Response:
The abbreviation, RCT, is now be expanded in its first appearance on page 4, line 153.
- Authors used many abbreviations that are not expanded, such as CSS (line 198), MDSCs (line 199) etc.
Response:
All abbreviations have now been double-checked through-out the manuscript to ensure that they are expanded when first introduced. The changes have been highlighted.
- In Figure 1A and B, authors obtained HR 0.72 in both cases. In fig 1B, the test of subgroups has a p value 0.21, signifies no difference. Whereas, in 1A, authors do not show any p-value (except heterogenicity) to compare or confirm. Why HR 0.72 should be favorable for G-CSF treatment? Is there any range of HR that should be considered as favorable or nonfavorable? These should be explained.
Response:
We thank the author for the questions. The stated p value in figure 1A is the p-value for heterogeneity in the meta-analysis, whereas the p-value of 0.21 is for the test for subgroup differences which can only be measured when subgroup analyses have been conducted. The HR value is below 1 (in this case 0.72) meaning that the measured outcome favors G-CSF treatment over placebo.
- Similarly in PFS, why HR 0.74 is not a significant improvement? Then authors should write no significant improvement in PFS over OS (line 233).
Response:
We are sorry for the confusion this may have caused. We have now described the results differently at page 8, line 245-247.
- Line 254, RR should be expanded.
Response:
The abbreviation, RR is expanded in its first appearance on page 4, line 148.
- Line 272, NOS should be expanded. RoB should be expanded (line 275)
Response:
The abbreviation NOS is expanded in its first appearance on page 4, line 157.
The abbreviation RoB is expanded in its first appearance on page 4, line 163.
Reviewer 4 Report
Comments and Suggestions for Authors
This is a systemic reviews, collected 13 articles from 2000 to 2022, for evaluation long term outcomes of Granulocyte Colony-Stimulating Factor (G-CSF) Treatment in gastrointestinal cancers. They found that G-CSF prophylaxis provides a reduction of neutropenia grade III/IV in patients with GI cancers and a favourable overall survival (OS), while progression-free survival (PFS) is unaffected.
However, I have some concerned
- There were different specific G-CSF drugs utilized in these studies, even 3 studies did not specify which G-CSF drug was used.
- The investigated 195 chemotherapy regimens were quite diverse and are visualized.
- No detail stage of different cancers were reported.
All of this data missing, may be difficult to evaluated OS and PFS.
Author Response
Reviewer 4:
This is a systemic reviews, collected 13 articles from 2000 to 2022, for evaluation long term outcomes of Granulocyte Colony-Stimulating Factor (G-CSF) Treatment in gastrointestinal cancers. They found that G-CSF prophylaxis provides a reduction of neutropenia grade III/IV in patients with GI cancers and a favourable overall survival (OS), while progression-free survival (PFS) is unaffected.
However, I have some concerned:
There were different specific G-CSF drugs utilized in these studies, even 3 studies did not specify which G-CSF drug was used.
The investigated 195 chemotherapy regimens were quite diverse and are visualized.
Response:
We thank the reviewer for the concerns raised. The use of different G-CSF drugs and different chemotherapy regimens in the included studies was among the limitations of this study. We have highlighted these limitations in the “Discussion” section on page 11, line 321-336.
No detail stage of different cancers were reported.
Response:
We thank the reviewer for the comment. All stages of cancer were included in the study. Specific stages of diseases were not considered exclusion criteria in the protocol. Unfortunately, consistent classification of cancer stages were lacking in the included studies and were therefore not registered in the subsequent characteristics and analysis.
All of this data missing, may be difficult to evaluated OS and PFS.
Response:
We have now ensured that the interpretation of the results is done in a more cautious manner. We have now rephrased the conclusion of this manuscript, highlighting the limitations and that the data should be interpreted with caution. The included sentences can be found on page 13, line 288-392.
Round 2
Reviewer 3 Report
Comments and Suggestions for Authors
The paper is improved.
1. The answers for comment 3 and 9 should be incorporated in the text of the manusxcript, so the readers would not have these questions.